# Surgery for an Uncommon Pathology: Pancreatic Metastases from Renal Cell Carcinoma—Indications, Type of Pancreatectomy, and Outcomes in a Single-Center Experience

**DOI:** 10.3390/medicina60122074

**Published:** 2024-12-17

**Authors:** Emil Matei, Silviu Ciurea, Vlad Herlea, Traian Dumitrascu, Catalin Vasilescu

**Affiliations:** 1Department of General Surgery, Fundeni Clinical Institute, Carol Davila University of Medicine and Pharmacy, Fundeni Street No. 258, 022328 Bucharest, Romania; emil.matei@drd.umfcd.ro (E.M.); silviu.ciurea@umfcd.ro (S.C.); catalin.vasilescu@umfcd.ro (C.V.); 2Department of Pathology, Fundeni Clinical Institute, Carol Davila University of Medicine and Pharmacy, Fundeni Street No. 258, 022328 Bucharest, Romania; vlad.herlea@umfcd.ro

**Keywords:** pancreatic metastases, renal cell carcinoma, pancreatectomy, morbidity, survival

## Abstract

*Background and Objectives*: The role of surgery in pancreatic metastases of renal cell carcinoma (PM_RCC) is highly controversial, particularly in the context of modern systemic therapies and the conflicting results of studies published so far. This study aims to explore a single surgical center experience (including mainly pancreatic resections) regarding the indications, the type of pancreatectomies, and early and long-term outcomes for PM_RCC. *Materials and Methods*: The data of all patients with surgery for PM_RCC (from 1 January 2002 to 31 December 2023) were retrospectively assessed, and potential predictors of survival were explored. *Results*: 20 patients underwent surgery for PM_RCC (pancreatectomies—95%). Metachronous PM_RCC was 90%, with a median interval between the initial nephrectomy and PM_RCC occurrence of 104 months. For elective pancreatectomies, the overall and severe morbidity and mortality rates were 24%, 12%, and 0%, respectively; 32% of patients underwent non-standardized pancreatic resections. The median survival of patients with negative resection margins was 128 months after pancreatectomies, with an 82% 5-year survival rate. Left kidney RCC and the body/tail PM_RCC were favorable prognostic factors for the overall survival after pancreatectomies for PM_RCC. Body/tail, asymptomatic PM_RCC, and an interval after initial nephrectomy > 2 were favorable prognostic factors for the overall survival after initial nephrectomy for RCC. *Conclusions*: Pancreatectomies for PM_RCC can achieve long-term survival whenever complete resection is feasible, with acceptable complication rates. Patients with left kidney RCC, body/tail, and asymptomatic PM_RCC and an interval of more than 2 years after nephrectomy exhibit the best survival rates.

## 1. Introduction

Pancreatic metastases (PM) of other neoplasms are an uncommon pathology, representing 0.5% to 5% of all malignant tumors of the pancreas [1,2,3,4], reaching 4–12% in the case of autopsies [5,6]. Renal cell carcinoma (RCC) is the most frequent primary source of PM; thus, pancreatic metastases of renal cancer origin (PM_RCC) represent 38.4% to 76.3% of all PM [7,8,9,10,11,12,13,14,15]. PM_RCC is encountered in the context of disseminated disease with other concomitant metastatic sites in most patients [7]. The role of surgery in PM_RCC with or without other extrapancreatic metastatic sites (even limited and potentially amenable for resection) is highly controversial [16]. However, in the few cases of isolated PM_RCC, there appears to be a rare particular biological behavior for these metastases, frequently solitary in the pancreas, with a slow growth pattern and a long time until development and appearance after RCC resection, compared with other metastatic sites in RCC. These features of isolated PM_RCC may make them more suitable for surgical resection [7,17,18,19,20,21,22].

Today, pancreatectomies in high-volume centers are associated with acceptable morbidity and mortality rates [23]. However, PM represents an exceptional indication for pancreatectomies even at high-volume centers of pancreatic resections [10,24].

The most significant part of single-center studies published so far reporting the outcomes of patients with pancreatectomies for PM_RCC included retrospective studies with a small number of patients (usually less than 20 patients) [13,24,25,26,27,28]. Thus, considering the relatively small number of surgically treated PM_RCC in previously published series [24,26,28], the advancement of modern therapies for metastatic RCC, including tyrosine kinase inhibitors (TKI) and immune checkpoint inhibitors (ICI) [16,29], and the conflicting results of studies published in the literature so far [16,27], the role of surgery is not well defined for PM_RCC. It is also worth mentioning that a few studies show significant heterogeneity in the immune response of PM_RCC, suggesting that this type of metastasis may exhibit resistance to immunotherapy [30].

The current study aims to add value to the current literature regarding the potential role and benefits of surgery for PM_RCC. It explores a single high-volume surgical center experience for surgery (including mainly pancreatic resections) regarding the indications, the type of surgical approach, and early and long-term outcomes for PM_RCC. Furthermore, potential predictors of disease-free and overall survival were explored.

## 2. Materials and Methods

The patients of the present series were retrospectively identified from a prospectively maintained electronic database of patients (including pancreatectomies) at the Department of General Surgery, Fundeni Clinical Institute, Bucharest, Romania, between January 2002 and December 2023. It is worth mentioning that during the abovementioned period, 2299 pancreatectomies for different indications were performed in our department.

Ethical approval was not required due to our internal institutional regulations for retrospective studies not interfering with patient management (PO-ADM-042/4 September 2023). The code states that for this type of research, ethical committee approval was not required.

The inclusion criteria were: (1) final pathological diagnosis of PM_RCC; (2) previous or concomitant diagnoses of fully resected RCC by nephrectomy; (3) surgical approach for PM_RCC; (4) available pre-, intra-, postoperative, and follow-up data. Exclusion criteria were: patients with surgery for (1) local invasion of the pancreas by the RCC; or (2) lymph node or local recurrence of the primary RCC. It is also worth mentioning the exclusion of two patients with pathological suspicion of pancreatic metastasis but without a prior or concomitant diagnosis of RCC; for these two particular patients, the further immunohistochemistry examination showed a final pathological diagnosis of a rare primary pancreatic tumor (i.e., clear cell adenocarcinoma).

Overall, 20 patients with surgery for PM_RCC and available data were identified and included in the present study.

Data collected from the medical records included age, gender, presenting symptoms, type of surgery/pancreatectomy, time from nephrectomy, number and location of PMs, presence of other metastatic sites, pathologic features, and postoperative course, including complications, recurrences, and overall survival. Follow-up data were obtained from the electronic data and medical records. All the patients were followed until death occurrence or last follow-up (i.e., 1 February 2024), mainly focusing on their oncological outcome.

A PM_RCC was defined as metachronous if developed six months after nephrectomy; otherwise, it was considered synchronous. All the patients were assessed by abdominal ultrasound and contrast-enhanced computed tomography; three underwent magnetic resonance imaging examination, and two underwent endoscopic ultrasound examination.

Postoperative morbidity and mortality were defined at 30 days from the surgical procedure for PM_RCC. All complications were identified and graded using the Clavien-Dindo classification, while the International Study Group of Pancreatic Surgery definition and grading were used for the postoperative pancreatic fistula.

### Statistical Analyses

Data are presented as median (range) for the continuous variables, except for the situations in which the median was not reached by Kaplan-Meier estimation (in this particular situation, mean ± SD was provided instead); data are presented as numbers (percentages) for the categorical variables. The Kaplan–Meier curves were plotted to determine disease-free survival after PM_RCC surgery and overall survival after nephrectomy and after surgery for PM_RCC; the log-rank test was used for the comparative analyses. The median follow-up time was calculated using the reversed Kaplan-Meier curves. Statistical analysis was performed using the “R” Program, version 4.4.0 Copyright (C)-The R Foundation for Statistical Computing, R Core Team (Vienna, Austria, 2024); probability values were considered statistically significant at *p* values less than 0.05.

## 3. Results

Overall, 20 patients with surgery for PM_RCC were identified: 6 females (30%) and 14 males (70%); the median age of patients in the present cohort was 64 years (range, 50–75 years). The medical history of the 20 patients with surgery for PM-RCC of the present series included a total right nephrectomy for 11 patients (55%) and a total left nephrectomy for the remaining 9 patients (45%). Histological examination of the nephrectomy specimen revealed a clear cell carcinoma in all cases.

The PM_RCC was synchronous in 2 patients (10%) and metachronous in 18 patients (90%). In patients with metachronous metastases, the median interval between the initial RCC nephrectomy and the development of PM_RCC was 104 months (range, 6–280 months). There were three patients (15%) with completely resected metastases occurring in other organs before the PM_RCC: one patient with lung metastasis, 2 years following nephrectomy and 3 years before the PM_RCC; one patient with liver metastasis, 16 years following nephrectomy and 7 years before the PM_RCC; one patient with mediastinal lymph node metastasis, 1 year following nephrectomy and 6 years before the PM_RCC.

Nine patients (45%) were asymptomatic, diagnosed by close oncological follow-up. The other 11 patients (55%) were symptomatic; the main complaints were abdominal pain (6 patients—30%), obstructive jaundice (3 patients—14%), and mild acute pancreatitis in one patient (5%). Two patients (10%) presented with massive upper digestive hemorrhage requiring emergency pancreatectomies.

The PM_RCC was localized in the head of the pancreas in 7 patients (35%), in the body/body—tail in 12 patients (60%), and in the pancreatic head and body in 1 patient (5%). The PM_RCC was singular in 16 patients (80%) and multiple in 4 patients (20%). Eight patients (40%) had synchronous extrapancreatic metastasis: liver (six patients—30%) and great omentum (two patients—10%).

Eighteen patients (90%) underwent elective surgical procedures. In comparison, two patients (10%) required surgery (i.e., pancreatectomy) in an emergency setting for massive, uncontrolled upper digestive hemorrhage (bleeding non-responsive to radiological embolization or endoscopic therapy). For the patients with elective surgery, 17 out of 18 patients (94.4%) underwent pancreatectomies, except for 1 patient who required a bilio-digestive anastomosis for obstructive jaundice (after failed endoscopic stenting).

Overall, 19 patients (95%) underwent pancreatectomies for PM_RCC. Thus, during the analyzed time, pancreatectomies for PM_RCC represented only 0.8% of the total number of pancreatectomies in our surgery department.

For the 19 patients with pancreatectomies for PM_RCC, the surgical procedures included a distal pancreatectomy in 11 patients (58%) (with splenectomy in 6 patients and with spleen preservation in 5 patients), pancreaticoduodenectomy in 5 patients (26%), total pancreatectomy in 2 patients (10%) (1 patient with spleen preservation), and enucleation in 1 patient (5%). All the surgical procedures were carried out using an open approach, except for one patient (5%), in which a laparoscopic distal spleno-pancreatectomy was performed. Thus, six patients underwent non-standardized pancreatic resections for PM_RCC (32%) (Table 1).

Six patients (32%) underwent additional surgical procedures for local invasion or removal of another metastatic site: liver resection (three patients—16%), partial omental resection (two patients—10%), and partial inferior vena cava resection (one patient—5%) (Table 1).

Negative resection margins were obtained in 17 patients (out of 19 patients with pancreatectomies)—89%, except for the 2 emergency palliative pancreatectomies for upper digestive hemorrhage, patients that presented multiple liver metastases.

Of the 19 patients with pancreatectomies for PM_RCC, 14 (74%) had an uneventful postoperative course. Thus, the overall morbidity rate in the cohort of patients with pancreatectomies for PM_RCC was 26%: grade I—one patient (5%), grade II—one patient (5%), grade III a—two patients (10%), and grad V—one patient (5%) (Table 1). The severe morbidity rate (i.e., grade III–V) was 15%. The main complications were grade B pancreatic fistula (three patients—16%) and peripancreatic abscesses (two patients—10%). For elective pancreatectomies, the overall and severe morbidity and mortality rates were 24%, 12%, and 0%, respectively. For the two patients with emergency pancreatectomy, the overall severe morbidity and mortality rates were 50% each. Thus, one patient with palliative pancreaticoduodenectomy for upper digestive hemorrhage in the context of multiple liver metastases died of multiple organ dysfunction secondary to uncontrolled abdominal sepsis (Table 1).

The patient with bilio-digestive anastomosis (not resected) died after 3 months due to the advanced disease.

Out of the 17 patients with pancreatectomies and negative resection margins, recurrence of the disease occurred in 5 patients (30%) during a median follow-up time after PM_RCC resection of 62 months (range, 9—128 months). The recurrence sites were pancreas only (two patients—12%), liver only (two patients—12%), and liver and pancreas in one patient (6%). An iterative pancreatectomy was possible in one patient with pancreatic-only recurrence, and the rest of the patients with recurrences were treated by chemotherapy.

The estimated median overall survival of 17 patients with pancreatectomies and negative resection margins for PM_RCC was 128 months (range, 9–128 months) after pancreatectomies (Figure 1A) and 283 months (range, 22–408 months) after initial nephrectomy for RCC (Figure 1B). The 1-, 5-, and 10-year estimated survival rates were 100%, 82%, and 60%, respectively, after pancreatectomies for PM_RCC and 100%, 100%, and 83%, after initial nephrectomy for RCC. The estimated mean disease-free survival after pancreatectomies for PM_RCC was 76 months (± 10.9), with 1-, 5-, and 10-year disease-free survival rates of 95%, 60%, and 60%, respectively (Figure 2).

The univariate analysis explored the following potential predictors for overall and disease-free survival after negative resection margins pancreatectomies for PM_RCC: age, gender, primary RCC location (left vs. right kidney), other previously resected metastases at another site, PM_RCC location (head vs. body/tail), other synchronous resected metastatic sites, unique vs. multiple PM_RCC, asymptomatic vs. symptomatic PM_RCC, the interval from initial nephrectomy, standard vs. non-standardized pancreatectomies, and the occurrence of postoperative complications. Thus, left kidney RCC origin (estimated median overall survival 128 vs. 60 months, *p* = 0.019) and PM_RCC in the body/tail (estimated median overall survival 128 vs. 60 months, *p* = 0.024) were identified as favorable prognostic factors for the overall survival after pancreatectomies for PM_RCC (Figure 3). PM_RCC in the body/tail (estimated median overall survival 283 vs. 120 months, *p* < 0.001), asymptomatic PM_RCC (estimated median overall survival 408 vs. 176 months, *p* = 0.029), and an interval after initial nephrectomy for RCC > 2 years (estimated median overall survival 283 vs. 67 months, *p* = 0.014) were identified as favorable prognostic factors for the overall survival after initial nephrectomy for RCC (Figure 4). None of the abovementioned factors were found to have a statistically significant impact on disease-free survival (*p* values ≥ 0.111).

## 4. Discussion

Today, there are many efficient therapeutic options for metastatic RCC, and systemic therapy plays a crucial role [7,14,16,29,31,32,33,34]. The treatment strategy is based on many factors, including the presence of solitary or multiple metastatic lesions/anatomical sites, the feasibility of complete resection, the presence of oligometastatic disease, and the patient’s biological condition; surgical metastasectomy remains a potential valid option in a subset of patients even though the use of modern targeted or immunotherapies represents the standard of care in metastatic RCC [18,29]. However, a few critical guidelines worldwide state the role of surgery in metastatic RCC differently. Thus, the National Comprehensive Cancer Network (NCCN) guidelines recommend resection of synchronous and metachronous metastases of RCC in patients with oligometastatic disease, particularly for metachronous metastases occurring a long time after radical nephrectomy and for those located in the lungs, bone, and brain; alternatives to surgery are considered local ablative therapy and stereotactic body radiotherapy [33]. The European Association of Urology guidelines recommend resectioning oligometastatic disease except for brain and bone metastases, where stereotactic therapy is considered therapeutical [31]. The American Society of Clinical Oncology (ASCO) guidelines recommend metastasis-directed therapy, including resection, ablative therapy, or radiotherapy for patients with low-volume metastatic RCC; surgery is suggested as the primary option in younger and fitter patients and for patients with solitary lung and adrenal metastasis. At the same time, in the ASCO guidelines, there is an emerging role of stereotactic ablative body therapy for oligometastatic RCC because of low morbidity and limited need to interrupt systemic therapy [32]. The European Society of Medical Oncology (ESMO) guidelines recommend local treatment of metastases of RCC whenever it is feasible, including surgery when a complete resection can be achieved, particularly for patients with good performance status, a solitary and metachronous pattern, long-term intervals after nephrectomy, and the absence of progression on systemic therapy [34].

Synchronous and metachronous metastases in RCC are encountered in 17–30% and 35–50% of patients, respectively [25,35]. The most frequent sites of metastases in RCC are the lungs, lymph nodes, bones, liver, brain, and adrenal glands [21,29,36]; about two-thirds of metastatic RCC patients present more than one metastatic site at diagnosis [37]. Thus, the pancreas remains an uncommon site for metastatic RCC [18,36]. The biology of RCC is heterogeneous concerning aggressivity and can explain the development of synchronous or metachronous PM_RCC. The most frequent PM_RCC are metachronous [5,38]. Synchronous PM_RCC are rare and associated with poor prognosis [39], explained by increased primary tumor aggressivity [7]. Most patients have metachronous patterns in the present cohort of surgically treated PM_RCC (90%). Studies that did not identify any differences in survival for resected synchronous vs. metachronous PM_RCC [40] are worth mentioning.

There are several theories for the occurrence of PM_RCC. The most plausible method of dissemination is hematogenous by selective nesting of embolized cells from the RCC in the pancreas—the “seed and soil” hypothesis [8,40,41]; however, this seeding mechanism does not entirely explain the appearance and development of isolated PM_RCC. Furthermore, it has been hypothesized that a selective affinity of RCC for the pancreas exists; there have been reports of PM_RCC occurring even in ectopic pancreatic tissue [42]. The fatty infiltration of the pancreas has been considered a risk factor for PM_RCC [43]. Other theories explaining the PM_RCC development, such as lymphatic, peritoneal, perineural, or continuous spreading of RCC, were not sufficiently documented [38,41,44,45].

The imagistic work-up is essential for the diagnostic and therapeutic management of PM_RCC. The initial assessment is standard/contrast abdominal ultrasound, followed by contrast-enhanced computed tomography, and certain cases requiring more refined methods such as magnetic resonance imaging or positron emission computed tomography. Positron emission computed tomography is particularly recommended in advanced metastatic stages of RCC, aiming to make a complete assessment and rule out extensive diseases. Since PM_RCC are hypervascular, differential diagnosis is mainly carried out with primary neuroendocrine tumors of the pancreas—a common misdiagnosis [46] (Figure 5)—but also with PMs originating from hepatocellular carcinomas or sarcomas, the final diagnosis being established by immunohistochemical examination. Endoscopic ultrasound is the gold standard approach in these patients, as it allows for guided fine needle aspiration with a definitive histopathological diagnosis. An accurate differential diagnosis is crucial because PM-RCC and neuroendocrine pancreatic tumors may have different approaches [46], including surveillance for small neuroendocrine tumors. Nevertheless, a study comparing the computed tomography features of PM_RCC and neuroendocrine tumors has shown that, compared to the PM_RCC, neuroendocrine tumors tend to be larger, more frequently solitary, and show calcifications and main pancreatic duct dilatation, being more heterogenous [47] (Figure 5). Interestingly, concomitant neuroendocrine and PM_RCC were described in the literature [48].

Several systematic reviews or meta-analyses assessing the role of metastasectomy on overall survival in metastatic RCC have associated it with improved survival compared with non-resected patients receiving systemic therapy or even local therapies, such as radiotherapy [7,13,14,26,27,35,36,37,38,49,50,51]. Complete resection, a solitary and metachronous pattern, a long-term interval after nephrectomy (more than 2 years), younger patients, and clear cell type RCC were identified as predictors of better outcomes. In contrast, non-lung metastasis, a synchronous pattern, multiple metastases/sites, high-grade tumors, and primary T-stage ≥ 3 were identified as predictors of poor outcomes [26,34,35,51,52]. It is widely accepted that metastatic sites are critical for the prognosis of patients with RCC, and lung metastases appear to have the best survival benefit from resection [26]. However, PM_RCCs have better long-term outcomes than other metastatic sites [6,10,13,14,17,19,27,37,50,53,54,55,56].

In the subgroup analyses of patients with isolated PM_RCC, Ouazid and coworkers have shown that patients with resection have had significantly better overall survival compared with the patients with only systemic therapy, with median overall survivals similar to those reported after resection of lung metastases; patients with solitary metastasis and long-term survival after nephrectomy are considered the best candidates for surgery [26].

Regarding the type of pancreatectomy for PM (including PM_RCC), it appears that non-standardized pancreatic resections (Figure 6) are a feasible and oncologically safe option [39,57,58], with acceptable morbidity and mortality rates, albeit with standard pancreatic resections being used in the most significant part of the patients [10,13,15,16,20,21,22,24,26,27,39,54,59,60]. An analysis of 414 patients with pancreatectomies for PM showed the use of pancreaticoduodenectomy, total pancreatectomy, distal pancreatectomy, and enucleation in 37.9%, 11.4%, 43.5%, and 7.2% of patients, respectively [15]. In this context, the reported morbidity and mortality rates after pancreatectomy for PM were 33–48.3% and 1.4–4.5%, respectively; the main complications were pancreatic fistula (9–47.6%), delayed gastric emptying (5.6–40.4%), and abscesses (6.3–18%) [10,15,20,22,27,28,60]. An analysis of 641 patients with pancreatectomies for PM_RCC showed the use of pancreaticoduodenectomy, total pancreatectomy, distal pancreatectomy, and enucleation in 30%, 18%, 39%, and 4% of patients, respectively; the severe morbidity and mortality rates were 25.4% and 4.2%, respectively [27]. Other studies reported overall and severe morbidity and mortality rates after pancreatectomy for PM_RCC between 19% and 64.3%, 14% and 21.4%, and 0–5.4%, respectively [20,22,24,26,60]. In the present cohort, the overall and severe morbidity and mortality rates after elective pancreatectomies for PM_RRC were 24%, 10%, and 0%, respectively, with POPF and peripancreatic abscesses as leading causes (Table 1).

A meta-analysis published in 2023, including 56 patients with enucleations for PM_RCC, showed overall and severe morbidity rates of 19.6% and 5.9%, respectively, with no mortality; at the same time, 5-year disease-free and overall survival rates of 79% and 82%, respectively, were reported (similar to those reported after standard pancreatic resections) [58]. However, Bassi and coworkers observed higher recurrence rates and decreased overall survival after non-standardized pancreatectomies for PM_RCC compared to the standard pancreatectomies [61]. As previous studies have shown [20,28,39,40,50,59], in the present cohort, no differences in disease-free and overall survival were observed between the patients with standard and non-standardized pancreatectomies for PM_RCC.

Nevertheless, it is worth mentioning that PM_RCC represents an uncommon indication for pancreatectomies even in high-volume centers of pancreatic resection (0.8% in our experience; 0.3% to 1.8% in other studies [9,43,62,63]; the morbidity and mortality rates are not higher than those reported for different indications [10,11].

There is a low rate of lymph node metastases in patients with standard pancreatectomies for PM_RCC—up to 23.8% of patients—with no significant impact on survival [15,17,20,45,54,56,60,64,65], except for two studies [39,66]. This feature might suggest parenchyma-sparing pancreatectomies over standard pancreatectomies for PM_RCC, aiming to preserve pancreatic functions better [22].

Synchronous resection of PM with other extrapancreatic metastases is reported in 19% to 41% of patients; interestingly, no negative impact on survival was observed when complete resection of extrapancreatic metastases was achieved along with pancreatectomies for PM [9,15,17,19,38,39,54,60,65,66,67,68]. Similar outcomes were observed in the present cohort of patients, where 30% of patients associated pancreatectomies for PM_RCC with resection of other metastatic sites, a situation that was not correlated with a negative impact on disease-free and overall survival.

The median and 5-year overall survival rates after pancreatectomies for isolated PM_RCC have been reported to be 53.7 to 120 months and 33% to 88%, respectively [7,9,11,12,13,15,16,17,18,21,24,25,26,27,28,38,39,40,45,54,55,56,59,60,62,65,66,68,69,70]. In contrast, systemic therapy only for PM_RCC is associated with a median and 5-year survival rate of 6 to 29 months and 10% to 47%, respectively [25,27,29,38,71,72]. In the present cohort, pancreatectomies for PM_RCC were associated with a median overall survival of 128 months after PM_RCC resection and an 82% 5-year survival rate.

Adjuvant therapy after complete resection of RCC metastases appears to have no benefit on long-term survival, except for pembrolizumab [52,73]. Current NCCN guidelines recommend adjuvant pembrolizumab after resection of metastatic RCC [33], while no systemic therapy is recommended in the ESMO guidelines [34].

A few potential predictors of long-term outcomes in patients with resected PM (including PM_RCC) should be discussed. It is widely accepted that patients with PM_RCC have longer survival rates than non-RCC patients [15,18].

The long-term disease-free interval between the nephrectomy for RCC and PM_RCC occurrence is considered a particular pattern [9,18,21]. It might be a surrogate for a less aggressive biological disease that warrants a pancreatectomy in PM_RCC [26]. The long period between RCC resection and the PM_RCC appearance can be explained by the initial equilibrium between dormant micrometastases and the immune system, which is disrupted by the activation of angiogenesis within the tumoral microenvironment [74] and the decline of immunity by immunosenescence and chronic inflammation [75]. Furthermore, recent studies have shown that PM_RCC presents unique immunological features, including a low intratumoral density of CD8+ and FOXP3+ lymphocytes and CD8+ T cells and fibroblasts as the most active incoming and outgoing components of the tumor microenvironment. A particular pattern of primary RCC appears to be correlated with PM_RCC occurrence [21]. These situations may reflect its less aggressive nature and potentially better prognosis than other metastatic sites [21,30,76]. A longer time to metastasis development after initial diagnosis of RCC is associated with improved survival, as an analysis of 7386 patients has shown [77], while other studies failed to show any impact on the survival of the interval between nephrectomy and PM_RCC occurrence [28,38,40,45,60,66]. A few studies have reported a median interval of 58 to 156 months between nephrectomy and PM_RCC occurrence [7,9,15,22,24,27,28,39,54,55,56,59,62,64,65,68,69,70]; the most significant part of studies reported intervals longer than 10 years [25]. Nevertheless, the most prolonged interval between nephrectomy and PM_RCC resection reported in the literature is 32.7 years [78]. In the present cohort, the median interval between nephrectomy for RCC and PM_RCC occurrence was 104 months (the largest being 280 months); an interval after initial nephrectomy for RCC > 2 years was associated with statistically significantly better overall survival after initial nephrectomy for RCC (Figure 4A), but no impact on disease-free survival or overall survival after PM_RCC was observed. Masetti and coworkers also associated at least 2 years after nephrectomy for RCCC with significantly better overall survival [67]. Considering the appearance of PM_RCC during a highly variable, sometimes long interval from the initial nephrectomy, these patients require a close, indefinite follow-up.

Approximately 50% of patients with PM are symptomatic at the time of diagnosis, with the most frequent symptoms and signs being abdominal pain (34.8%), jaundice (20.6%), and upper digestive hemorrhage (9.2%) [15,27]. Thus, many patients with PM_RCCC are diagnosed during the oncological follow-up [9]. Symptomatic patients with PM have worse survival rates than those without symptoms at diagnosis [14,15,38,56,67], as in the present cohort (Figure 4C), 55% were symptomatic. Other studies failed to show any impact of symptoms on overall survival in PM_RCC [18,54].

A few patients in the present cohort (10%) presented with massive upper digestive hemorrhage that required emergency palliative pancreaticoduodenectomy, a situation that was associated with high morbidity and mortality (Table 1). This complication, more frequent than in primary malignant tumors of the pancreas, can be explained by the hypervascular character of the PM_RCC. Life-threatening complications of PM_RCC, such as upper gastrointestinal hemorrhage not responsive to radiological embolization, require extensive surgical resection, even if associated with extrapancreatic metastases, with increased morbidity and mortality. Emergency pancreatectomies for non-traumatic indications are widely accepted to have higher morbidity and mortality rates compared with elective ones, with up to 34% mortality rates [79,80].

Positive resection margins are a poor prognostic factor after PM resection, including PM_RCC [15,37,54,67]. However, negative resection margins after pancreatectomies for PM_RCC are reported in 84% to 100% of patients [15,20,22,28,55,65]. In the present cohort, the rate of negative resection margins was 89% (except for the two emergency palliative pancreatectomies for upper digestive hemorrhage, patients that presented multiple liver metastases). A few studies showed no impact of margin status on overall survival in PM_RCC [45,62,66,81].

There is no correlation between the site of RCC (left vs. right kidney) and the development of PM_RCC within the pancreas [41,44,45,60,81], also demonstrated in this series. However, in the present study, left kidney RCC was associated with statistically significantly better overall survival after pancreatectomy for PM_RCC (Figure 3A), while no impact was observed on disease-free survival. Other studies failed to show any effect on the survival of the primary tumor location [60].

In the present study, PM_RCC of the body and tail of the pancreas had statistically better overall survival after both initial nephrectomy for RCC and pancreatectomy for PM_RCC (Figure 3B and Figure 4B), compared with PM_RCC in the pancreatic head. However, no impact on disease-free survival was observed. Other studies failed to show any prognostic value for PM_RCC location [60,67,81].

A few studies analyzing prognostic factors affecting survival after pancreatectomies for PM_RCC found no impact of single vs. multiple pancreatic lesions [9,17,27,38,39,40,41,60,62,66,67,82], as was the case in the present study. However, extrapancreatic metastases were associated with poorer survival [13,14,27,38,56], a situation not confirmed in the present study’s results.

The recurrence rate after pancreatectomies for PM_RCC was frequent—35.2% to 71.6% [9,17,18,20,22,24,39,55,56,60,65,68,69,70], and it was associated with symptomatology, number, and size of PM_RCC, extrapancreatic disease, and limited pancreatic resection [38,60,61]. Thus, a few studies reported an increased recurrence rate in patients with standard pancreatic resection [58], explained by the imbalance between the host immunological reaction and a slow-growing tumor. The recurrence rate in the present cohort was 30%, but no predictors of disease-free survival were identified, as a recent study has also shown [28]. The reported median disease-free survival in previously published studies varies between 22 and 107.5 months [17,27,28,39,56].

Recurrences after pancreatectomies for PM_RCC can involve the pancreatic stump (pancreatic recurrence—the most frequent site) and/or develop at a distance (extrapancreatic recurrence) and are usually treated by systemic therapy. In surgically fit patients (considering the age and comorbidities) with pancreatic recurrence, iterative pancreatectomies can be performed [17,20,22,24,60,70]. In this series, only one patient underwent an iterative pancreatectomy.

The results of the present study (Table 2) should be interpreted with caution because there are several limitations: the retrospective design and low number of patients with pancreatectomies for PM_RCC could bias the statistical comparative analyses, the long-analyzed interval that may imply changes of surgical and non-surgical local and systemic therapies (particularly the introduction of TKI and ICI therapies), and no comparative analyses to alternative local or systemic treatments such as local ablation and radiotherapy, TKI, and ICI. Thus, a few studies found no significant long-term outcome differences for PM_RCC surgically treated or with systemic therapies [16,53,83,84]. Furthermore, a few studies have shown the best responses to targeted therapy for PM_RCC compared with other metastatic sites [19,71,85,86]. In contrast, other studies suggested the lack of clinical benefits of targeted therapies [87] or ICI in PM_RCC [19,88]. Encouraging oncological results were also reported with local ablative therapies for PM_RCC [89]. Nevertheless, a recent multicentric study confirmed that patients with PM_RCC have more prolonged overall survival even in the context of other metastatic sites with systemic therapies; no significant differences in overall survival were observed between patients with surgery and systemic therapies for oligometastatic PM_RCC [16].

## 5. Conclusions

PM_RCC appears to have a particular pattern that might make it more suitable for surgical resection: a slow-growing pattern and long interval occurrence after RCC resection. Pancreatectomies for PM_RCC can be safely performed and should be offered to surgically fit patients whenever negative resection margins are anticipated, especially those with metachronous patterns, being associated with high disease-free and overall survival rates. Parenchyma-sparing pancreatectomies should be the first option in patients with PM_RCC and an indication for surgery, without jeopardizing the oncological safety. Patients with left kidney RCC, body/tail, asymptomatic PM_RCC, and an interval of more than 2 years after nephrectomy exhibit the best survival rates.

## Figures and Tables

**Figure 1 medicina-60-02074-f001:**
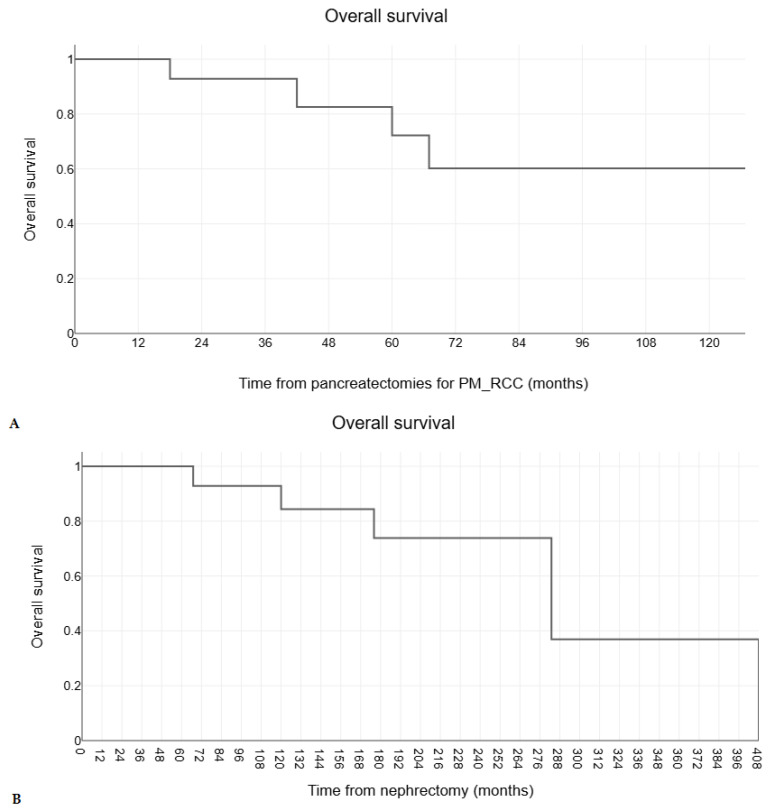
The Kaplan-Meier survival curves for overall survival (**A**) after pancreatectomies for PM_RCC and (**B**) after initial nephrectomy for RCC in 17 patients with negative resection margins pancreatectomies for PM_RCC.

**Figure 2 medicina-60-02074-f002:**
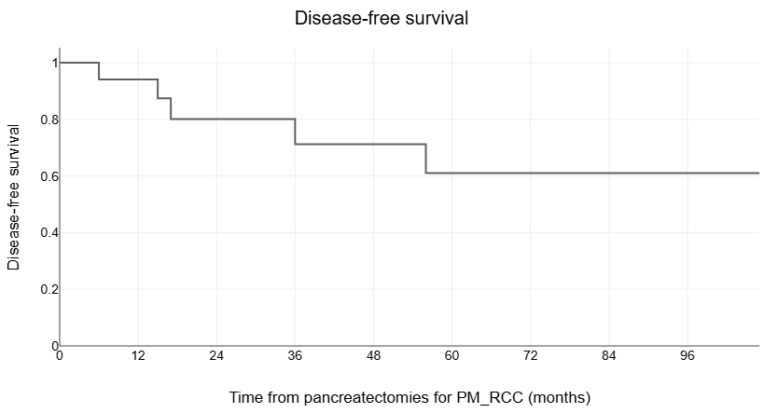
The Kaplan-Meier survival curves for disease-free survival in 17 patients with negative resection margins pancreatectomies for PM_RCC.

**Figure 3 medicina-60-02074-f003:**
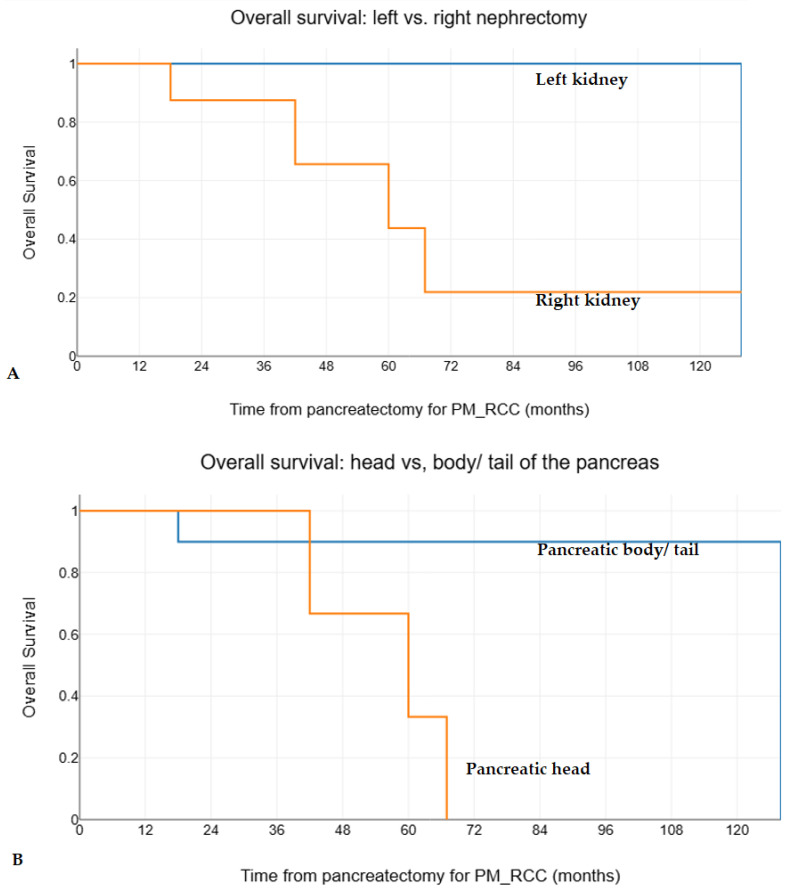
The comparative Kaplan-Meier survival curves for overall survival after pancreatectomy in 17 patients with negative resection margins pancreatectomies for PM_RCC, stratified by (**A**) initial RCC location (left vs. right kidney, *p*-value = 0.019) and (**B**) PM_RCC location (pancreatic head vs. body/tail, *p*-value = 0.024).

**Figure 4 medicina-60-02074-f004:**
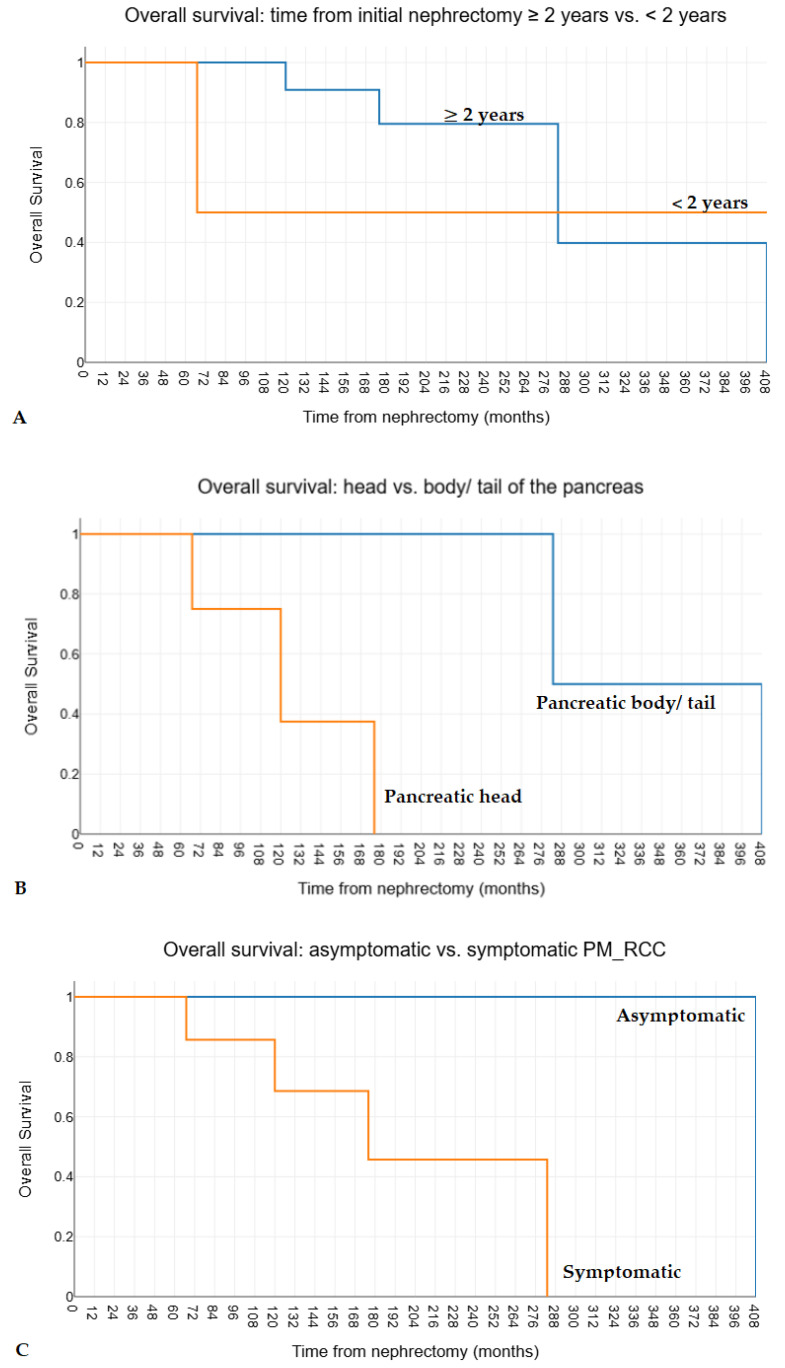
The comparative Kaplan-Meier survival curves for overall survival after initial nephrectomy for RCC in 17 patients with negative resection margins pancreatectomies for PM_RCC, stratified by (**A**) interval from initial nephrectomy (>2 years vs. ≤2 years, *p*-value = 0.014), (**B**) PM_RCC location (pancreatic head vs. body/tail, *p*-value < 0.001), and (**C**) presence of symptoms (asymptomatic vs. symptomatic, *p*-value = 0.029).

**Figure 5 medicina-60-02074-f005:**
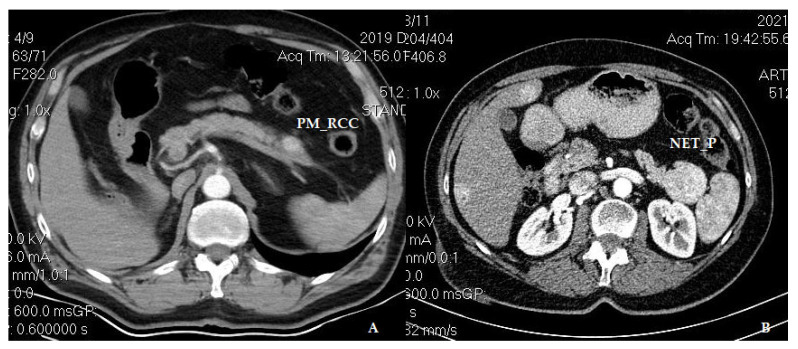
In the tail of the pancreas, a hypervascular appearance of PM_RCC (**A**) and NET_P (**B**) at contrast-enhanced axial computed tomography, arterial phase (PM_RCC—pancreatic metastasis of renal cell carcinoma origin; NET_P—neuroendocrine tumor of the pancreas).

**Figure 6 medicina-60-02074-f006:**
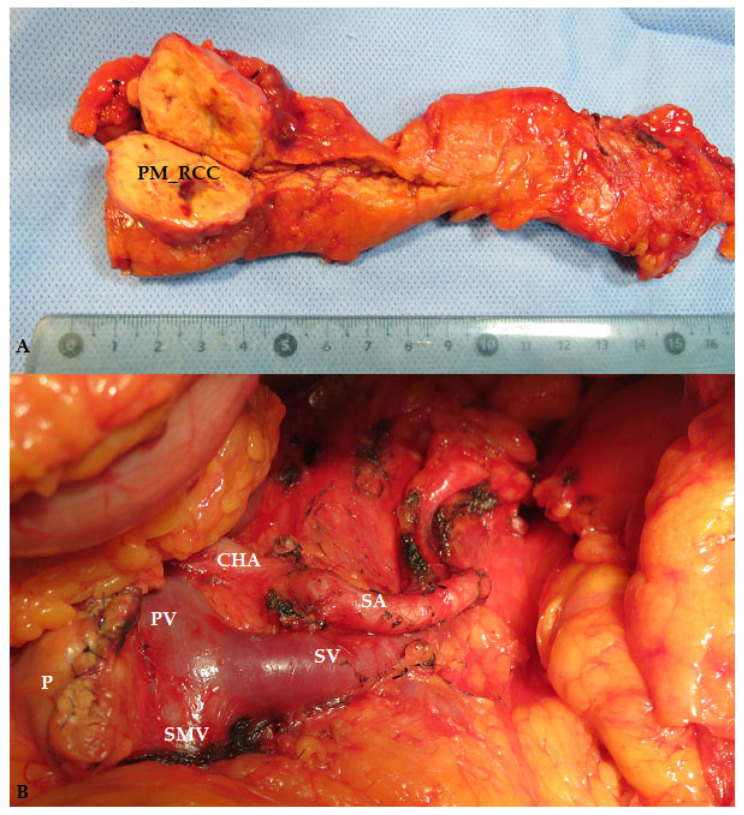
Operative specimen (**A**) and intraoperative aspects (**B**) after spleen-preserving distal pancreatectomy for PM_RCC (PM_RCC—pancreatic metastasis of renal cell carcinoma origin; P—pancreatic head; CHA—common hepatic artery; SA—spleen artery; PV—portal vein; SMV—superior mesenteric vein; SV—spleen vein).

**Table 1 medicina-60-02074-t001:** Types of resections and early morbidity in 19 patients with pancreatectomies for PM_RCC.

Type of Pancreatic Resection	No of pts	Early Morbidity	POPF Grade B	Peripancreatic Abscesses	Mortality
Spleen-preserving distal Pancreatectomy ^1^	5 pts	2 pts (40%)	2 (40%)	-	-
Distal spleno-pancreatectomy ^2^	6 pts	-	-	-	-
Pancreaticoduodenectomy ^3,^*	5 pts	2 pts (40%)	1 pt (20%)	1 pt (20%)^*^	1 pt (20%) *
Spleen-preserving total pancreatectomy	1 pt	-	-	-	-
Total pancreatectomy with splenectomy *	1 pt	-	-	-	-
Enucleation ^4^	1 pt	1 pt (100%)	-	1 pt (100%)	-
Total	19 pts	5 pts (26%)	3 pts (16%)	2 pts (10%)	1 pt (5%)

POPF–postoperative pancreatic fistula; ^1^ one pt with atypical liver resection of segment V and one patient with partial omentectomy; ^2^ one pt with laparoscopic approach and one patient with atypical liver resection of segment VI and VII; ^3^ one pt with partial omentectomy and one pt with inferior vena cava lateral resection; ^4^ one pt with atypical liver resection of segment 2 and 3; * emergency pancreatectomy for upper digestive surgery.

**Table 2 medicina-60-02074-t002:** Outcomes of pancreatectomies for PM_RCC in recent studies.

Author, Year	Center	No of pts	Period	Median Interval from Nephrectomy	Morbidity Rate *	Median Follow-Up Time *	Recurrence Rate *	5-Year DFS *	Median OS *	5-Year OS *
Di Franco G et al., 2019 [17]	University Hospital of Pisa	21	1999–2019	83 months	8.6%	77 months ¥	42.9%	-	75 months	71.6%
Patyutko Y et al., 2019 [59]	Blokhin National Cancer Medical Research Center, Vishnevsky National Medical Research Center of Surgery	54	1995–2018	120 months	52%	73 months	-	31%	84 months	74%
Bauschke A et al., 2019 [55]	University of Jena	19	1995–2018	156 months	31% #	-	43%	-	69 months	73%
Anderson B et al., 2020 [69]	Barnes-Jewish Hospital and the Alvin J. Siteman Cancer Center, Washington University School of Medicine	29	1995–2017	96 months	44.8%	64 months	55.2%	45%	-	-
Brozzetti S et al., 2020 [65]	University of Rome La Sapienza	26	2002–2015	156 months	53.8%	104 months ¥	46.2%	57.7%	-	76.9%
Chikhladze S et al., 2020 [9]	University of Freiburg Medical Centre	20	2005–2017	116 months	70%	76.4 months	70%	56%	54 months	89%
Milanetto A et al., 2020 [39]	Multicentric, 3 hospitals, Italy	39	2000–2019	84 months	38.5%	68 months	48.7%	60%	134 months	79%
Malleo G et al., 2021 [20]	University of Verona, Memorial Sloan Kettering Cancer Center	69	2000–2008	109 months	34.8%	141 months	71.6%	-	-	-
Blanco-Fernandez G et al., 2022 [60]	Multicentric, 40 hospitals, Spain	116	2010–2020	87.3 months	60.9%	43 months	53.4%	35%	105 months	83%
Cignoli D et al., 2022 [21]	Multicentric, 3 hospitals, Italy	33	1987–2018	96 months	-	101 months	-	-	106 months	75.1%
Moletta L et al., 2023 [56]	University of Padua	16	2000–2019	111 months	43.7% #	33 months	56.2%	31%	119 months	72%
Boubaddi M et al., 2024 [22]	Bordeaux University Hospital	42	2005–2022	121 months	23.8% #	76 months	69%	29.6%	-	92.8%
Hajibandeh S et al., 2024 [28]	University Hospitals of North Midlands NHS Trust	18	2008–2021	58 months	61.1%	-	-	55.5%	64 months	55.5%
Al-Madhi S et al., 2024 [70]	University Hospital of Magdeburg	17	2010–2022	154 months	65%	43 months	35.2%	70%	-	72%
Riemenschneider K et al., 2024 [68]	Rigshospitalet Copenhagen	25	2011–2021	95.6 months	36%	69.6 months	60%	32.3%	46.3 months	83.6%
Present series	Fundeni Clinical Institute Bucharest	17	2002–2023	104 months	24%	62 months	30%	60%	128 months	82%

OS—overall survival; DFS—disease-free survival; * after pancreatectomy for PM_RCC; # severe morbidity; ¥ mean.

## Data Availability

The datasets generated during and/or analyzed during the current study are available from the corresponding author upon reasonable request.

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
