# Peer review of "Surgery for an Uncommon Pathology: Pancreatic Metastases from Renal Cell Carcinoma—Indications, Type of Pancreatectomy, and Outcomes in a Single-Center Experience"

_medicina, 2024, doi:10.3390/medicina60122074_

Round 1
Reviewer 1 Report
Comments and Suggestions for Authors
The authors have presented an interesting manuscript regarding the clinical-surgical implications of patients with pancreatic metastases from renal clear cell carcinoma. The work has been well set up, the references of comparison with other studies regarding highly specialized centers for these pathologies in Europe and in the world, are appropriate. The work overall is good and brings a further contribution on this type of pancreatic resections. However, we would like to further encourage the authors to increase the iconographic aspect of the manuscript. The addition of histological and immunohistochemical images of RCC metastases would improve the visual aspect of the manuscript. Likewise, the authors could insert the color graphs (blue and yellow lines) related to the KM curves.
Author Response
Reviewer 1
“The authors have presented an interesting manuscript regarding the clinical-surgical implications of patients with pancreatic metastases from renal clear cell carcinoma. The work has been well set up, the references of comparison with other studies regarding highly specialized centers for these pathologies in Europe and in the world, are appropriate. The work overall is good and brings a further contribution on this type of pancreatic resections.”
Comment 1. “However, we would like to further encourage the authors to increase the iconographic aspect of the manuscript. The addition of histological and immunohistochemical images of RCC metastases would improve the visual aspect of the manuscript.”
Response to reviewer comment 1: Thank you for your comment, suggestions, and kind appreciation of the present manuscript. The authors have in preparation a paper addressing the differential pathologic diagnosis of renal cell carcinoma metastasis to the pancreas and clear cell carcinoma of the pancreas. Thus, we would prefer no to provide in the present manuscript any histological or immunohistochemical images. Thanks for your understanding.
Comment 2. “Likewise, the authors could insert the color graphs (blue and yellow lines) related to the KM curves.”
Response to reviewer comment 2: The authors have considered the reviewer's suggestion and modified Figures 3 and 4.
Reviewer 2 Report
Comments and Suggestions for Authors
Matei et al present a single center experience of 20 cases of RCC with subsequent resection of pancreatic metastases. The presented series is of high interest. A couple of considerations:
Table 1: Consider writing the type of pancreatic resektion in the table instead of abbrieviations for increased readability.
Line 378-381 seems off topic in the discussion since the scope is beyond adjuvant treatmernt.
Line 469 Consider change to should be "interpreted" with caution
Consider removing table 2, more suitable for a narrative review article
Conclusion: Only include conclusions drawn from your own research, i.e. shorten.
Author Response
Reviewer 2
“Matei et al present a single center experience of 20 cases of RCC with subsequent resection of pancreatic metastases. The presented series is of high interest. A couple of considerations:”
Comment 1. “Table 1: Consider writing the type of pancreatic resection in the table instead of abbreviations for increased readability.”
Response to reviewer comment 1: The authors have considered the reviewer's suggestion and modified Table 1.
Comment 2. “Line 378-381 seems off topic in the discussion since the scope is beyond adjuvant treatment.”
Response to reviewer comment 2: The impact of adjuvant therapy on long-term outcomes was not analyzed in the present cohort because no reliable data about this aspect were available. However, nowadays, adjuvant therapy is recommended in a few important guidelines. That is the reason why we have introduced this statement.
Comment 3. “Line 469 Consider change to should be "interpreted" with caution”
Response to reviewer comment 3: The authors have considered the reviewer's suggestion and modified the text.
Comment 4. “Consider removing table 2, more suitable for a narrative review article.”
Response to reviewer comment 4: Table 2 shows that even studies published in recent years include a relatively low number of patients worldwide. Furthermore, a comparative analysis would be beneficial. Nevertheless, Table 2 could be switched to Supplementary Material, as the journal staff has the latitude to do so.
Comment 5. “Conclusion: Only include conclusions drawn from your own research, i.e. shorten.”
Response to reviewer comment 5: The authors have considered the reviewer's suggestion and shortened the conclusions to include only those emerging from the present study's results.
Round 2
Reviewer 1 Report
Comments and Suggestions for Authors
We would like to thanks the authors to make the changes suggested.